# Discovery of Novel Thiosemicarbazides Containing 1,3,5-Triazines Derivatives as Potential Synergists against Fluconazole-Resistant *Candida albicans*

**DOI:** 10.3390/pharmaceutics14112334

**Published:** 2022-10-29

**Authors:** Fei Xie, Yumeng Hao, Jiacun Liu, Junhe Bao, Tingjunhong Ni, Yu Liu, Xiaochen Chi, Ting Wang, Shichong Yu, Yongsheng Jin, Liping Li, Dazhi Zhang, Lan Yan

**Affiliations:** 1Department of Organic Chemistry, School of Pharmacy, Naval Medical University, No. 325 Guohe Road, Shanghai 200433, China; 2Center of New Drug Research, School of Pharmacy, Naval Medical University, No. 325 Guohe Road, Shanghai 200433, China; 3Department of Pharmacy, Shanghai Tenth People’s Hospital, School of Medicine, Tongji University, No. 1239 Siping Road, Shanghai 200092, China; 4School of Traditional Chinese Materia Medica, Shenyang Pharmaceutical University, No. 103 Wenhua Road, Shenyang 110016, China

**Keywords:** 1,3,5-triazines, drug resistance, antifungal activity, synergistic, synthesis

## Abstract

The clinical prevalence of antifungal drug resistance has been increasing over recent years, resulting in the failure of treatments. In an attempt to overcome this critical problem, we sought novel synergistic enhancers to restore the effectiveness of fluconazole against resistant *Candida albicans*. Based on the structural optimization of hit compound **8** from our in-house library, a series of novel 1,3,5-triazines derivatives was designed, synthesized, and biologically evaluated for synergistic activity in combination with fluconazole. Among them, compounds **10a**–**o**, which contain thiosemicarbazides side chains, exhibited excellent in vitro synergistic antifungal potency (MIC_80_ = 0.125–2.0 μg/mL, FICI range from 0.127 to 0.25). Interestingly, compound **10l** exhibited moderate *C. albicans* activity as monotherapy with an MIC_80_ value of 4.0 μg/mL, and also on several *Cryptococcus* strains (MIC_80_ ranging from ≤ 0.125–0.5 μg/mL) and *C. glabrata* (MIC_80_ ≤ 0.125 μg/mL). These effects were fungal-selective, with much lower levels of cytotoxicity towards human umbilical vein endothelial cells. Here, we report a series of thiosemicarbazides containing 1,3,5-triazines derivatives as potent synergists with fluconazole, and have preliminarily validated compound **10l** as a promising antifungal lead for further investigation.

## 1. Introduction

Due to advances in modern medicine, antifungal drug resistance has become a widespread clinical problem that is frequently encountered. The associated morbidity and mortality rates of invasive fungal infections (IFIs) have increased dramatically in recent years [1,2]. *Candida albicans* is the most common opportunistic fungi, with mortality rates of 40–80% [3,4,5]. Moreover, IFIs generally occur in patients with critical comorbidities, such as viral pneumonitis (including SARS-CoV-2), and those living in an immunocompromised state, for example those living with AIDS or immunosuppressive therapy following organ transplant [6,7]. Currently, there are four major classes of antifungal drugs: polyenes, echinocandins, flucytosine, and azoles. Fluconazole is a widely available drug worldwide and has high bioavailability, good water solubility, low toxicity, and is relatively inexpensive. However, fluconazole has frequently been associated with the development of drug resistance. Current research to optimize the azoles, either through the development of a new generation of azoles (e.g., the tetrazoles) or use in combination therapy, has shown significant clinical results.

The drug combination has proven an effective strategy to overcome drug resistance in *C. albicans* [8]. Several small molecules have no or weak antifungal activity by themselves yet have been identified as successfully restoring the antifungal activity of fluconazole against resistant fungi when the two drugs are used in combination. These small molecules are regarded as enhancers or synergists. Our group had reported multiple studies on the use of drug combination, such as baicalein [9], curcumin [10], berberine [11], and a series of structural optimization of berberine [12,13,14]. Here, we have continued to persist with our search to discover new types of synergistic molecules.

Due to the prevalence of electronegative nitrogen atoms in polyazine systems, triazine, as an electron-deficient aromatic scaffold, has higher nucleophilic substitution/addition reactivity compared to benzene. Numerous routes have been proposed for the synthesis of triazine and its derivatives exhibiting various physical, chemical, and biological properties. These could be potential building blocks and start-ups in drug design and discovery research [15]. As an important class of the heterocyclic systems, 1,3,5-triazines (s-triazine) have been extensively studied in medicinal chemistry (Figure 1) on account of their pharmacological properties: being anticancer [16], antiviral [17], antibacterial [18], antimalarial [19] and anti-inflammatory [20]. 1,3,5-Triazines and their derivatives have also been reported to possess antifungal activity [21,22,23,24]; however, to the best of our knowledge, 1,3,5-triazines or their derivatives have not been used as antifungal synergists to restore the susceptibility of the antifungal triazoles to azole-resistant fungi.

In this study, we designed, synthesized, and evaluated the antifungal activity of four series of trisubstituted 1,3,5-triazines derivatives, which was based on the screening of our hit compound **8** from an in-house library (Figure 2). The PEG-containing side chains in hit compound **8** had no drug-like properties. Firstly, we replaced the PEG-containing side chains and retained the other two substituted groups, (4-fluorophenyl)methanamine and cyclohexanemethylamine. Then, by retrosynthetic analysis, we synthesized two key intermediates, acid **5** and hydrazide **6**, to further react with various aromatic amines, isocyanate, and isothiocyanate, respectively. Through optimizing side chains, we obtained 42 target compounds in a facile way. Among them, the series compounds **10a**–**o** containing a thiosemicarbazides moiety exhibited excellent synergists activity with fluconazole against fluconazole-resistant *C. albicans*. Compound **10l** also possessed moderate activity against fluconazole-resistant *C. albicans* by itself, with selective anti-*Cryptococcus* and anti-*C*. *glabrata* activity. Overall, based on a hit-to-lead structural optimization, a class of novel thiosemicarbazides containing 1,3,5-triazines derivatives with excellent synergist activity were discovered in vitro.

## 2. Experimental Section

### 2.1. Chemistry

^1^H and ^13^C NMR spectra were recorded in DMSO-*d_6_* unless otherwise indicated with a Bruker AC-300P spectrometer, using tetramethylsilane (TMS) as an internal standard. Chemical shifts (δ values) and coupling constants (*J* values) are given in ppm and Hz, respectively. ESI mass spectra were performed on an Agilent Technologies 6120 Quadrupole LC-MS. TLC analysis was carried out on silica gel plates GF254 (Yantai Huanghai Chemical, Qingdao, China). The solvents and reagents were purchased from commercial vendors and were used as received, or dried prior to use as needed.

#### 2.1.1. General Procedure for Synthesis of Methyl 2-(4-((4,6-Dichloro-1,3,5-triazin-2-yl)amino)phenyl)acetate (**2**)

Methyl 2-(4-aminophenyl)acetate (2.34 mL, 16.3 mmol) was added dropwise to a solution of cyanuric chloride (**1**) (3.00 g, 16.3 mmol) and NaHCO_3_ (1.36 g, 16.3 mmol) in THF (50 mL) in an ice-water bath. The solution was stirred at 0 °C for 3 h and then concentrated in vacuo. The mixture was recrystallized by MeOH/H_2_O, filtered, and washed with water to afford intermediate **2** as a white solid (4.21 g, 83%).

#### 2.1.2. General Procedure for Synthesis of Methyl 2-(4-((4-Chloro-6-((4-fluorobenzyl)amino)-1,3,5-triazin-2-yl)amino)phenyl)acetate (**3**)

(4-fluorophenyl)methanamine (1.54 mL, 13.4 mmol) was added dropwise to a solution of intermediate **2** (4.20 g, 13.4 mmol) and NaHCO_3_ (1.13 g, 13.4 mmol) in THF (50 mL). The reaction was stirred at room temperature for 2 h. After the reaction was completed (monitored by TLC with PE/EtOAc = 4/1, *v*/*v*), the mixture was concentrated in vacuo and then extracted with ethyl acetate (3 × 30 mL). The organic phases were combined, washed with brine (2 × 50 mL), and dried over anhydrous Na_2_SO_4_. The crude product was purified by chromatography on silica gel (PE/EtOAc = 10:1~1:1) to obtain intermediate **3** as a white solid (3.67 g, 68%).

#### 2.1.3. General Procedure for Synthesis of Methyl 2-(4-((4-((Cyclohexylmethyl)amino)-6-((4-fluorobenzyl)amino)-1,3,5-triazin-2-yl)amino)phenyl)acetate (**4**)

1,8-diazabicyclo [5.4.0]undec-7-ene (DBU, 2.23 mL, 14.96 mmol) was added to a mixture of disubstituted triazine **3** (3.00 g, 7.48 mmol) and cyclohexylmethanamine (1.26 mL, 9.72 mmol) in 1,4-dioxane (50 mL). The reaction was stirred at reflux conditions overnight. After the reaction was completed (monitored by TLC with DCM/MeOH = 15/1, *v*/*v*), the mixture was concentrated in vacuo and then extracted with ethyl acetate (3 × 30 mL). The organic phases were combined, washed with brine (2 × 50 mL), and dried over anhydrous Na_2_SO_4_. The crude product was purified by chromatography on silica gel (DCM/MeOH = 150:1~80:1) to obtain ester **4** as a semisolid (2.50 g, 70%).

Compound **4**: ^1^H NMR (300 MHz, DMSO-*d_6_*) δ 9.05–8.73 (m, 1H), 7.68 (d, *J* = 8.5 Hz, 2H), 7.46–7.02 (m, 7H), 6.84 (t, *J* = 32.8 Hz, 1H), 4.44 (d, *J* = 5.7 Hz, 2H), 3.60 (s, 3H), 3.57 (s, 2H), 3.10–3.02 (m, 2H), 1.73–1. 40 (m, 6H), 1.23–0.99 (m, 3H), 0.99–0.68 (m, 2H). ^13^C NMR (75 MHz, DMSO-*d_6_*) δ 172.35, 166.43, 166.30, 166.18, 164.58, 163.08, 159.88, 139.98, 137.61, 137.33, 129.50, 129.42, 129.31, 127.25, 120.09, 119.97, 119.82, 115.38, 115.10, 99.99, 52.02, 46.98, 46.66, 43.27, 43.15, 42.98, 38.08, 37.92, 31.07, 26.65, 25.93. MS (ESI) *m*/*z*: 479.0 [M + H]^+^, C_26_H_31_FN_6_O_2_. Purity: 97.6% (LC-MS).

#### 2.1.4. General Procedure for Synthesis of 2-(4-((4-((Cyclohexylmethyl)amino)-6-((4-fluorobenzyl)amino)-1,3,5-triazin-2-yl)amino)phenyl)acetic acid (**5**)

NaOH (2 mol/L, 20 mL) was added to a solution of ester **4** (2.00 g, 4.18 mmol) in MeOH (30 mL). The reaction was stirred at reflux for 5 h (monitored by TLC, DCM/MeOH = 20/1, *v*/*v*). The organic solvent was removed in vacuo and the residues were dissolved in water. The mixture was acidified with 2 mol/L HCl until the product precipitated. The precipitate was filtered and dried to afford acid **5** as a white solid (1.04 g, 54%).

Compound **5**: ^1^H NMR (300 MHz, DMSO-*d_6_*) δ 12.20 (s, 1H), 9.03–8.63 (m, 1H), 7.75–7.53 (m, 2H), 7.46–7.01 (m, 7H), 6.83 (t, *J* = 30.7 Hz, 1H), 4.43 (d, *J* = 5.0 Hz, 2H), 3.45 (s, 2H), 3.10–3.02 (m, 2H), 1.74–1.36 (m, 6H), 1.23–0.97 (m, 3H), 0.95–0.61 (m, 2H). ^13^C NMR (75 MHz, DMSO-*d_6_*) δ 173.48, 166.39, 166.33, 166.28, 166.21, 166.17, 164.57, 163.09, 159.86, 139.74, 137.35, 129.52, 128.02, 120.11, 119.75, 115.39, 115.11, 46.99, 46.66, 43.23, 43.12, 43.04, 42.96, 37.95, 31.08, 26.65, 25.94. MS (ESI) *m*/*z*: 465.0 [M + H]^+^, C_25_H_29_FN_6_O_2_. Purity: 96.9% (LC-MS).

#### 2.1.5. General Procedure for Synthesis of 2-(4-((4-((Cyclohexylmethyl)amino)-6-((4-fluorobenzyl)amino)-1,3,5-triazin-2-yl)amino)phenyl)acetohydrazide (**6**)

NH_2_-NH_2_.H_2_O (4 mL, 83.6 mmol) was added to a solution of ester **4** (2.00 g, 4.18 mmol) in MeOH (30 mL). The reaction was stirred at reflux overnight (monitored by TLC, DCM/MeOH = 15/1, *v*/*v*). The solution was restored to room temperature and poured onto ice water (20 mL). The precipitate was formed, filtered, and recrystallized by MeOH/H_2_O twice to obtain hydrazide **6** as a white solid (1.24 g, 62%).

Compound **6**: ^1^H NMR (300 MHz, DMSO-*d_6_*) δ 9.15 (s, 1H), 8.95–8.63 (m, 1H), 7.74–7.52 (m, 2H), 7.42–7.07 (m, 7H), 6.84 (t, *J* = 34.0 Hz, 1H), 4.44 (d, *J* = 5.4 Hz, 2H), 4.23 (s, 2H), 3.25 (s, 2H), 3.15–2.95 (m, 2H), 1.73–1.46 (m, 6H), 1.19–1.03 (m, 3H), 0.99–0.68 (m, 2H). ^13^C NMR (75 MHz, DMSO-*d_6_*) δ 170.37, 166.42, 166.29, 166.19, 164.56, 163.06, 159.93, 159.86, 139.52, 137.39, 137.34, 129.41, 129.27, 129.12, 120.12, 119.69, 115.39, 115.11, 46.93, 46.61, 43.26, 37.95, 31.08, 26.65, 25.93. MS (ESI) *m*/*z*: 479.0 [M + H]^+^, C_25_H_31_FN_8_O. Purity: 98.2% (LC-MS).

#### 2.1.6. General Procedure for Synthesis of Target Compounds (**8** and **8a**–**j**)

Benzotriazol-1-yl-oxytripyrrolidinophosphonium hexafluorophosphate (PyBOP, 0.17 g, 0.33 mmol) and *N*, *N*-diisopropylethylamine (DIEA, 156 μL, 0.90 mmol) in DMF (5 mL) was added to the corresponding amine (0.30 mmol), to a solution of acid 5 (0.14 g, 0.30 mmol) and the mixture was stirred at room temperature for 3 h (monitored by TLC, DCM/MeOH = 20/1, *v*/*v*). The reaction solution was diluted with water (10 mL) and filtered. The precipitate was recrystallized by MeOH/H_2_O to obtain the target compound (**8a**–**j**).

To remove the Boc-protecting group, HCl (5.0 mL, 4.0 M in EtOAc) was added dropwise into a solution of tert-butyl (2-(2-(2-(2-(4-((4-((cyclohexylmethyl)amino)-6-((4-fluorobenzyl)amino)-1,3,5-triazin-2-yl)amino)phenyl)acetamido)ethoxy)ethoxy)ethyl)carbamate (**8a**) (0.15 g, 0.22 mmol) in EtOAc (5.0 mL), cooled by an ice-water bath. The ice bath was removed, and the mixture was stirred at room temperature over a 1 h period. After completion of the reaction (monitored by TLC, DCM/MeOH = 10/1, *v*/*v*), the precipitate was filtered quickly and washed with dry ethyl ether to afford the hit compound **8** (86.0 mg, 67%) as a white salt for store. In this paper, we used it as a free amido form.

Compound **8**: ^1^H NMR (300 MHz, DMSO-*d_6_*) δ 10.43 (d, *J* = 105.8 Hz, 1H), 9.31–8.63 (m, 2H), 8.28–8.08 (m, 3H), 7.71–6.93 (m, 8H), 4.65–4.39 (m, 2H), 3.64–3.57 (m, 2H), 3.56–3.42 (m, 8H), 3.23–3.12 (m, 4H), 2.94 (dd, *J* = 10.5, 5.2 Hz, 2H), 1.81–1.44 (m, 6H), 1.30–1.02 (m, 3H), 1.03–0.71 (m, 2H). ^13^C NMR (75 MHz, DMSO-*d_6_*) δ 171.06, 166.41, 166.30, 164.55, 163.06, 159.85, 139.50, 137.35, 137.30, 129.43, 129.30, 129.12, 120.11, 119.86, 119.68, 115.39, 115.11, 70.41, 70.06, 69.97, 69.55, 46.93, 46.63, 43.26, 42.19, 37.96, 31.06, 26.65, 25.92. MS (ESI) *m*/*z*: 595.0 [M + H]^+^, C_31_H_43_FN_8_O_3_. Purity: 98.6% (LC-MS).

#### 2.1.7. General Procedure for Synthesis of Target Compounds (**9a**–**n**)

The corresponding isocyanates (0.32 mmol) was added to a solution of hydrazide 6 (0.14 g, 0.30 mmol) in anhydrous THF (5 mL). The mixture was then stirred at room temperature for 2 h (monitored by TLC, DCM/MeOH = 10/1, *v*/*v*). The organic solvent was removed in vacuo and the residues were recrystallized by MeOH/H_2_O to give the target compounds (**9a**–**n**).

#### 2.1.8. General Procedure for Synthesis of Target Compounds (**10a**–**o**)

The corresponding isothiocyanates (0.32 mmol) was added to a solution of hydrazide **6** (0.14 g, 0.30 mmol) in anhydrous EtOH (5 mL). The mixture was then stirred at reflux for 2 h (monitored by TLC, DCM/MeOH = 10/1, *v*/*v*). The organic solvent was removed in vacuo and the residues were recrystallized by MeOH/H_2_O to give target compounds (**10a**–**o**).

#### 2.1.9. General Procedure for Synthesis of Target Compounds (**11c**, **11i** and **11j**)

A volume of 0.30 mmol of compound **10c**, **10i** and **10j** and 0.90 mmol of *N*-(3-dimethylaminopropyl)-*N′*-ethylcarbodiimide hydrochloride (EDC·HCl) with catalytic amounts of hydroxybenzotriazole (HOBt) was added to 5 mL of anhydrous DMF. The reaction mixture was stirred at room temperature for 3–5 h under nitrogen atmospheric conditions. After the reaction was completed (monitored by TLC, DCM/MeOH = 15/1, *v*/*v*), the mixture was poured onto ice. The precipitate was filtered, washed with water, dried under high vacuum, and recrystallized by MeOH/H_2_O to give target compounds (**11c**, **11i** and **11j**).

### 2.2. Biological Activity

#### 2.2.1. Strains and Culture Conditions

The fungi strains used in this study were obtained from the American-Type Culture Collection (ATCC) and Changhai Hospital of Shanghai, China. All the strains were stored in glycerol and grown in yeast extract–peptone–dextrose (YPD, 1% yeast extract, 2% peptone, and 2% dextrose) medium at 30^◦^C with shaking (200 rpm) for 16 h. All the target compounds were prepared as 6.4 mg/mL solutions with DMSO.

#### 2.2.2. Antifungal Susceptibility Testing

The in vitro MIC_80_ values of the compounds against the fungi strains were determined using the microbroth dilution method, according to the protocols from CLSI document M27-A3. Briefly, fungal cells were resuspended in a RPMI 1640 medium and adjusted to 1 × 10^3^ cells/mL. Antifungal compounds were serially 2-fold diluted, with concentrations ranging from 64.0 to 0.125 µg/mL. The final concentration of fluconazole was 8.0 μg/mL for the preliminary screening assay. For the checkerboard microdilution, the final concentration of fluconazole was set in the range of 16.0–0.5 μg/mL. The plates were incubated at 30 °C for 48 h for *C. albicans*, *C. parasilosis*, *C. glabrata*, *C. tropicalis*, and *C. auris*, and 72 h for *C. neoformans* and *Aspergillus fumigates*. The optical density at 630 nM (OD_630_ nm) of each well was measured by spectrophotometer.

#### 2.2.3. Biofilm Formation Assay

Biofilm formation was performed according to the reported protocol [25]. *C. albicans* SC5314 strain was incubated to the exponential growth stage in YPD medium, then harvested and diluted in the RPMI 1640 medium at a concentration of 1 × 10^6^ CFU/mL. The fungal cells were transferred to the 96-well culture plates and static cultured at 37 °C. After 1.5 h of adhesion, the cells were washed three times with phosphate-buffered saline (PBS) and the non-adherent cells were removed. Different concentrations of FCZ and compounds were added, and the cells were further incubated at 37 °C for 24 h. Finally, the XTT/menadione reduction assay was used to examine the formation of biofilms. The optical density at 492 nM (OD_492_ nm) in each well was measured by a spectrophotometer. The assay was performed in triplicate. The comparison between the two group was determined by Student’s *t* test.

#### 2.2.4. Cytotoxicity Assays

The cytotoxicity of target compounds on human umbilical vein endothelial cell line (HUVECs) was determined according to previous work [26]. HUVECs were cultured in DMEM medium to 2 × 10^5^ cells/mL. A volume 100 μL of cell suspension was plated in 96-well plates. After adhesion for 12 h, the cell supernatant was replaced with 100 μL fresh DMEM medium containing different concentrations of the tested compounds dissolved in DMSO. Subsequently, the HUVECs were cultured for an additional 24 h at 37 °C with 5% CO_2_. Then, 10 μL of CCK-8 regents was added to the 96-well plates and the HUVECs were incubated for another 2 h. The cell survival percentage was assessed by detecting absorbance at 450 nm with a microplate reader.

## 3. Results and Discussion

### 3.1. Chemistry

The complete synthesis routes of target compounds are described in Figure 1, Figure 2, Figure 3 and Figure 4.

2,4,6-Trichloro-1,3,5-triazine (TCT, **1**) is the starting material. The first nucleophile performs at 0 °C, while the second requires room temperature, and the third performs under reflux conditions. By carefully controlling the temperature stepwise in this way, we facilitated the consecutive nucleophilic substitutions with corresponding original aniline, benzylamine, primary amine in reaction activity order from low to high. Finally, we obtained the pure intermediate ester **4** with a relatively high yield, 70%. After regular hydrolysis and hydrazinolysis, the key intermediate acid **5** and hydrazide **6** were obtained, respectively (Figure 1).

Subsequently, acid **5** and various aromatic amines underwent a condensation reaction using PyBOP as a coupling reagent to give series **8b–j**. In this method, we also synthesized compound **8a**, then cleaved the Boc protecting group with hydrochloric acid to create hit compound **8** as a control (Figure 2).

The series **9a–n** and **10a–o** were prepared according to the procedures shown in Figure 3. Hydrazide **6** reacted with various commercially available isocyanates and isothiocyanates by a nucleophilic addition to produce the corresponding semicarbazides and thiosemicarbazides moiety-containing target compounds.

EDC-catalyzed cyclization of part thiosemicarbazides of compounds **10c**, **10i** and **10j** yielded the target compounds **11c**, **11i** and **11j,** as illustrated in Figure 4.

### 3.2. In Vitro Antifungal Evaluation and Structure–Activity Relationships Study

The in vitro synergistic antifungal activities of newly synthesized 1,3,5-triazine derivatives were tested with the broth microdilution susceptibility assay, guided by Clinical Laboratory Standards Institute (CLSI, USA) in CLSI document *M38eA2* [27]. The individual MIC_80_ values were obtained by determining the minimum concentration of the target compound, or of fluconazole used alone, required to inhibit ≥80% growth of fungal cells compared to the drug-free control. The interaction MIC_80_ values were determined for wells containing a combination of target compounds and fluconazole. Two fluconazole-resistant *C. albicans* strains, 901 and 904, were chosen for testing the activity of drug combinations. For determining the interaction modes of 1,3,5-triazine derivatives with fluconazole as synergistic or indifferent, the fractional inhibitory concentration index (FICI) was defined according to FICI values of ≤0.5 or >0.5, respectively, and calculated by summing up the ratios of the MIC_80_ (combination)/MIC_80_ (alone) of each compound and fluconazole^14^. It should be noted that, when the MIC_80_ value was >64.0 μg/mL, 64.0 μg/mL was used in the calculation.

As shown in Table 1, hit compound **8** in combination with fluconazole had moderate activity against the fluconazole-resistant *C. albicans* 901 and 904, with an MIC_80_ = 32.0 μg/mL, and FICI values of 0.625 and 0.375, respectively. These results indicated it was necessary for the structure to be optimized, and that SAR studies would be a useful way to obtain new potential synergists based on compound **8**. Thus, we displaced the PEG-containing amphiprotic side chains and kept the amide by coupling acid **5** with a series of amines, such as -F, -CH_3_, -OCH_3_, -OCH_2_CH_2_OCH_3_, substituents on the phenyl, and benzyl, furan and phenylfuran in series **8b**–**j**. However, none of the amide compounds were synergistic in combination with fluconazole (FICI > 1). Replacing the long PEG-containing side chains with various aromatic amines had no effects on optimizing hit compound **8**.

Next, we adjusted the side chains, and made use of NH_2_-NH_2_.H_2_O by hydrazinolysis of ester compound **4** to obtain hydrazide intermediate **6**. Intermediate **6** was then reacted with various aromatic isocyanate or isothiocyanate to generate compounds **9a**–**n** and compounds **10a**–**o**. Surprisingly for two structurally similar series of compounds, **9a**–**n** and **10a**–**o** exhibited completely opposite results. All of the semicarbazide compounds **9a**–**n** were ineffective when used alone or in combination with fluconazole against azole-resistant *C. albicans*, irrespective of the substituent on the terminal phenyl ring. In contrast, all thiosemicarbazides compounds **10a**–**o** showed excellent synergistic activity. For *C. albicans* 901 and 904, the interaction MIC_80_ values of **10a**–**o** were in the range of 0.125–2.0 μg/mL and the FICI values were 0.127–0.25, indicating that these compounds had a higher level of synergistic activity than hit compound **8**. Moreover, it seemed that compounds **10a–m** (R^3^ = Cl, F, CH_3_, CF_3_, OCF_3_, CN, NO_2_) which had a substituted group on the terminal phenyl, were more active compared to compound **10n** (R^3^ = H) against *C. albicans* 901, apart from compound **10g** (R^3^ = 2-CH_3_). The MIC_80_ of compound **10o** was 2.0 μg/mL against isolates *C. albicans* 901 and 904, which was due to the introduction of a benzyl group causing decreased synergistic activity compared with phenyl isothiocyanate. Compound **10m** (R^3^ = 4-NO_2_), which bears a strong electron withdrawing group, was the most active compound in our study, with the interaction of MIC_80_ 0.125 μg/mL against *C. albicans* 901 and 904. In addition, compound **10l** (R^3^ = 4-CN) unexpectedly displayed moderate activity against resistant *C. albicans*, with an MIC_80_ of 4.0 μg/mL.

Interestingly, as we introduced 1,3,4-oxadiazole moieties by cyclization of thiosemicarbazides (**10c**, **10i**, and **10j**) to afford the target compounds **11c**, **11i**, and **11j**, the synergistic activities were totally lost. It was clear that the thiosemicarbazides moiety was the key pharmacophore and played an important role in exerting synergistic antifungal activity in these trisubstituted 1,3,5-triazines derivatives.

### 3.3. Checkerboard Microdilution Assay

During the preliminary screening with 8.0 μg/mL FCZ, compounds **10a****–o** displayed good in vitro synergistic antifungal activity. Thus, four representative active compounds **10a**, **10j**, **10l**, and **10m** were selected for a checkerboard microdilution assay. A series of FICI and interaction MIC_80_ values were determined for these compounds, and the lowest FICI value is shown in Table 2. At concentrations of 32.0–0.125 μg/mL, the MIC_80_ of tested compounds decreased when combined with fluconazole at concentrations 16.0–0.5 μg/mL, as can be seen in the Appendix A. The lowest FICI values of compounds **10a**, **10j**, **10l**, and **10m** were 0.023, 0.039, 0.141, and 0.016, respectively, and showed superior synergistic activity to compound **8** (0.508). The lowest MIC_80_ values, with fluconazole in the parentheses, were 0.5(1.0), 0.5(2.0), 0.5(1.0) and 0.5(0.5) μg/mL. Overall, compounds **10a**, **10j**, **10l** and **10m** at 0.5 μg/mL decreased the MIC_80_ of fluconazole from >64.0 μg/mL to 0.5–2.0 μg/mL.

In summary, the above results further confirmed that the thiosemicarbazides containing 1,3,5-triazines derivatives possess synergistic activity with fluconazole. Furthermore, it is worthwhile noting that the higher FICI values of **10l** (0.141) compared to **10a**, **10j**, and **10m** (0.023, 0.039, 0.016) were due to the moderate antifungal activity of compound **10l** alone against the fluconazole-resistant *C. albicans*. Interestingly, Verma AK et al. recently reported a molecular docking and molecular dynamics modeling to investigate the interaction between the derivatives of 1,2,4-triazine with the enzyme lanosterol 14-demethylase (CYP51) of *C. albicans*, a key enzyme in sterol biosynthesis and the target of azoles antifungal drugs. They concluded that 1,2,4-triazine and its derivatives can target CYP51, which may be beneficial for bringing into antifungal activity [28]. These findings inspire us to further investigate the mechanism of compound **10l**.

### 3.4. Antifungal Spectrum Investigation of Compound **10l**

Inspired by the antifungal activity of compound **101** against the FCZ-resistant *C. albicans*, we further investigated the in vitro susceptibility of **10l** as a single agent against other pathogenic fungi including drug-resistant and -sensitive *C. albicans*, *Cryptococcus neoformans*, *Candida parapsilosis*, *Candida glabrata*, *Candida tropicalis*, *Candida auris* and *Aspergillus fumigates* (Table 3). Hit compound **8** showed broad-spectrum moderate antifungal activity (MIC_80_ 16.0–32.0 μg/mL) against these strains. Although compound **10l** had poor antifungal activity against many of the tested strains, it exhibited excellent and selective anti-*Cryptococcus* and anti-*C*. *glabrata* activity with an MIC_80_ below 0.125 μg/mL, lower than FCZ. To confirm the specific inhibitory effects of compound **10l** against *Cryptococcus*, a further 8 clinical *C. neoformans* fungal pathogens were assayed. The results listed in Appendix A indicate that **10l** did have potent antifungal activity against multiple strains of *Cryptococcus* (MIC_80_ range: ≤0.125–0.5 μg/mL), except for *C. neoformans* HN-15. Therefore, **10l** demonstrated potential as a novel antifungal agent for the treatment of cryptococcal meningitis.

### 3.5. Inhibitory Effects of Compounds **10l** and **10m** against the Formation of C. albicans Biofilms

Biofilms, an important virulence factor in C. albicans, are highly resistant to several clinical antifungal agents [29]. There is an intimate relationship between fungal biofilm formation and drug resistance. It was reported that the MIC values of fluconazole against different C. albicans strains under biofilm growth conditions were 60–4000 times higher than those against planktonic cells [30]. To further investigate the synergism of potent compounds **10l** or **10m** with FCZ on biofilm formation, we conducted a biofilm inhibitory assay using the 2,3-bis-(2-methoxy-4-nitro-5-sulfophenyl)-2H-tetrazolium-5-carboxanilide salt (XTT) reduction method. Compounds **10l** and **10m**, used alone, did not inhibit biofilm formation in the range of 0.0625–1 μg/mL. When the concentration was above 2 μg/mL, both compounds exhibited weaker inhibitory effects on fungal biofilm formation. As a control, FCZ at concentrations of 2, 4, or 8 μg/mL blocked nearly 92% of biofilm formation, and when the concentration decreased to 0.0625 μg/mL, it showed a 37% inhibition rate (Figure 3A). In particular, the inhibitory activity of FCZ at 0.0625 μg/mL on biofilm formation was improved by over 65%, and 62% when it was used in combination with compound **10l** or **10m** at a lower concentration of 0.0625 μg/mL, respectively (Figure 3B). However, when the FCZ was used at 0.5 or 8 μg/mL, our compounds showed a limited synergism with FCZ, mainly due to the potent inhibitory effects of FCZ itself.

### 3.6. In Vitro Cytotoxicity Assay

Due to the new types of triazine synergists, we examined whether the antifungal activity was caused by non-specific cytotoxicity. The cytotoxicity of active compounds **10a**, **10j**, and **10l** were tested in a human umbilical vein endothelial cell line (HUVECs) by the CCK8 method. As shown in Figure 4, when compounds **10a** and **10l** were used alone at 32 μg/mL and 64 μg/mL, they displayed a slight inhibitory effect on the HUVEC cells (survival rate means: 83.19% and 81.88%, respectively), and at 4 μg/mL there was greater than 90% survival. In contrast, compound **10j** showed moderate inhibitory effects on the HUVEC cells at 32 μg/mL (survival rate means: 65.28%), but greater than 90% survival at 4 μg/mL. Overall, the above results demonstrated that these compounds have a good safety profile at much lower MIC levels.

### 3.7. In Silico Predictive Studies

Drug-likeness is a significant drug research and development parameter that promotes molecules to be potential drug candidates. Therefore, it is necessary to evaluate these newly synthesized compounds’ ability as drugs by in silico predictions. As shown in the bioavailability radar formed by the SwissADME [31] (Figure 5), the physicochemical characteristics, lipophilicity, water solubility, and drug-likeness properties of four active compounds **10a**, **10j**, **10l**, and **10m** were all outside in the pink area (optimal range), indicating that these compounds had poor physicochemical properties and drug-likeness, such as size bulkiness (molecular weight > 500 Da), high polarity (TPSA > 130 Å^2^), poor water solubility (Log S (ESOL) < −6.00) and being too flexible (more than 9 rotatable bonds). Although these triazine derivatives exhibited excellent synergistic antifungal effects, they still had a lot of room for us to ameliorate their drug ability through further structural modifications.

## 4. Conclusions

In summary, we described the process of structural optimization of hit compound **8** with a focus on its side chains. A series of novel 1,3,5-triazines derivatives were designed, synthesized, and evaluated their potential as novel synergistic enhancers. Preliminary structure–activity relationship studies revealed that, when the PEG-containing side chains in compound **8** were replaced with the thiosemicarbazides moiety, the synergistic activity of compounds **10a**–**o** was maintained and further improved (MIC_80_ = 0.125–2.0 μg/mL, FICI range from 0.127 to 0.25) in combination with fluconazole. **10a**, **10j**, **10l**, and **10m** were the four potent compounds in combination with fluconazole out of the **10a**–**o** series, and so were tested by checkerboard microdilution assays. Results indicated that low concentrations (0.5 μg/mL) of these compounds could restore fluconazole from MIC_80_ > 64.0 μg/mL to 0.5–2.0 μg/mL, which in other words significantly sensitized fungi towards fluconazole. To our delight, compound **10l** also exhibited moderate activity against resistant Candida (MIC_80_ = 4.0 μg/mL) and selective anti-Cryptococcus and anti-C. glabrata activity (MIC_80_ ≤ 0.125 μg/mL). **10l** is a promising lead for the development of novel antifungal agents based on the new structure scaffold of the triazine nucleus. Overall, thiosemicarbazides containing 1,3,5-triazines derivatives (compounds **10a**–**o**) possess excellent synergistic antifungal activity, and low levels of cytotoxicity; however, these derivatives had poor physicochemical properties and drug-likeness. Therefore, further structure optimizations, focusing on improving drug-likeness and preliminary mechanism studies to confirm therapeutic targets, are currently underway.

## Data Availability

Not applicable.

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
