# Peer review of "Discovery of Novel Thiosemicarbazides Containing 1,3,5-Triazines Derivatives as Potential Synergists against Fluconazole-Resistant Candida albicans"

_pharmaceutics, 2022, doi:10.3390/pharmaceutics14112334_

Round 1

Reviewer 1 Report

In the manuscript by Xie et al., a novel class of chemical compounds were synthesized and demonstrated to have substantial antifungal activity against several clinically relevant Candida spp. Notably, the authors highlighted the physiochemical limitations associated with the identified antifungal compounds to be adopted as prospective drug candidates for antifungal interventions. Nevertheless, the efficacy of antifungals to treat candidiasis is limited; thus, information presented in this study has potential value to pharmacologists and dental communities. 

Author Response

Thank you for your kind suggestions and comments! Please find the attached file with our point-by-point response. 

Reviewer 2 Report

The present manuscript aimed to design, synthesize, and biologically evaluate several compounds for their synergistic activity in combination with fluconazole against many strains of candida. To achieve this result, the author designed and synthesized about 44 compounds and determined their MIC and FICI values against Candida albicans. The authors also determined the cytotoxicity of compounds with excellent antimicrobial properties. As a result, one compound stood out. Compound 10l exhibited moderate C. albicans activity as monotherapy with an MIC80 value of 4.0 μg/mL and also on several Cryptococcus strains (MIC80 ranging from ≤ 0.125−0.5 μg/mL) and C. glabrata (MIC80 ≤ 0.125 μg/mL). These effects were fungal-selective, with much lower levels of cytotoxicity towards human umbilical vein endothelial cells.

This reviewer is unfamiliar with all the chemical procedures to obtain the compounds. However,  my research field evaluates the biological properties of different compounds or substances. The MIC and FICI are the preliminary assays when assessing the antimicrobial properties of novel compounds. This method is suitable for screening several distinct compounds, as the authors performed in the present manuscript. However, MIC is too simple to support the future possible use of these compounds. In this manner, my only suggestion is to include further experiments analyzing the compounds over candida biofilms. It is well-known in the literature that microorganisms organize themselves as biofilms in the human body. Therefore, when analyzing novel substances, a biofilm assay is essential.

Primary strengths: number of compounds synthesized and analyzed and novelty of the objective

Primary weaks: absence of biofilm assay with the compound with the best MIC and FICI values

As a minor suggestion, please remove the following part from Conclusion: “these derivatives had poor physicochemical properties and drug-likeness, such as size bulkiness (molecular weight > 500 Da), high polarity (TPSA > 130 Å 2 ), poor water solubility (Log S (ESOL) < -6.00) and too flexible (more than 9 rotatable bonds), predicted by the SwissADME [25].”

This text is more appropriate for the Discussion section.

Author Response

(The authors gave the same response as above.)

Reviewer 3 Report

Identification of 1, 2, 4-Triazine and Its Derivatives Against Lanosterol 14-Demethylase (CYP51) Property of Candida albicans: Influence on the Development of New Antifungal Therapeutic Strategies.

1. The statement from Introduction: "1,3,5-triazines and its derivatives have also been reported to have antifungal activity [20-23], albeit much less relative to other biological activities." is not entirely true. Some of the 1,3,5-trizine derivatives described in ref. 20 demonstrated promising antifungal in vitro activity. More importantly, all triazine derivatives presented in ref. 20-23 are very different from the ones described in this ms. They share the same heterocyclic core but it is the only similarity. In consequence their molecular targets could be quite different, so that a direct comparison is not entirely justified.

2. The recent review: Mondal J., Sivaramakrishna A., Functionalized Triazines and Tetrazines: Synthesis and Applications. Topics in Current Chemistry, 380, 34 (2022) should be addressed in Introduction.

3. I would suggest addressing: Verma AK et al., Identification of 1, 2, 4-Triazine and Its Derivatives Against Lanosterol 14-Demethylase (CYP51) Property of Candida albicans: Influence on the Development of New Antifungal Therapeutic Strategies. Front. Med. Technol. 28, 845322 (2022) in Discussion

4. Quite many language problems (mainly grammar). The text should be corrected by any native English person.

Author Response

(The authors gave the same response as above.)

Round 2

Reviewer 2 Report

The authors answered all of my questions.  I believe the manuscript is ready for publication